# Automated Control of Rehabilitation Process in Physical Therapy Using a Novel Human Skeleton-Based Balanced Time Warping Algorithm

**DOI:** 10.3390/s25216696

**Published:** 2025-11-02

**Authors:** Oleg Seredin, Andrey Kopylov, Egor Surkov, Nikita Mityugov, Alexei Tokarev, Parama Bagchi, Debotosh Bhattacharjee

**Affiliations:** 1Laboratory of Cognitive Technologies and Simulation Systems, Tula State University, Lenin Ave. 92, 300012 Tula, Russia; oseredin@yandex.ru (O.S.); eg-su@mail.ru (E.S.); nikita.mityugov.2001@mail.ru (N.M.); 2Medical Institute, Tula State University, Lenin Ave. 92, 300012 Tula, Russia; mr.tokarev71@yandex.ru; 3Department of CSE, RCC Institute of Information Technology, Beliaghata, Kolkata 700015, India; paramabagchi@gmail.com; 4Department of CSE, Jadavpur University, Kolkata 700032, India; debotosh@ieee.org

**Keywords:** rehabilitation assessment, skeletal models, motion analysis, weighted Euclidean distance, time warping

## Abstract

Physical therapy is a critical component of medical rehabilitation, aiding recovery from conditions such as stroke, spinal cord injuries, and musculoskeletal disorders. Effective rehabilitation requires precise monitoring of patient performance to ensure exercises are executed correctly and progress is accurately assessed. This paper presents a novel automated system for controlling the rehabilitation process and evaluating physical therapy exercise quality using computer vision and a customized Human Skeleton-based Balanced Time Warping algorithm. The proposed method quantitatively assesses the similarity between a physiotherapist and patient performance by analyzing skeletal motion data extracted from RGB-D video sequences without requiring pre-alignment or sensor-specific calibration. A motion-dependent, weighted Euclidean distance between 3D skeletal models is used to compute pose dissimilarity, while a modified DTW approach aligns temporal sequences and evaluates dynamic consistency. The total dissimilarity measure is a balanced combination of posture (DP) and dynamics (DT) components. Evaluated on a custom dataset of 136 video recordings from 23 participants performing exercises in sitting and standing positions under varying performance accuracy levels (“good,” “intermediate,” and “bad”), the system demonstrates the strong clustering of accuracy levels. Proposed dissimilarity, together with a fixed reference element (physiotherapist), induces a natural non-strict order on the set of distances between patients and physiotherapists. A high value of Spearman’s rank correlation coefficient between computed dissimilarity and execution accuracy (0.977) indicates that this method is suitable for assessing exercise performance accuracy and for adequately evaluating the patient’s rehabilitation progress. The method enables objective, real-time feedback, reduces therapist workload, and supports remote monitoring, offering a scalable solution for personalized rehabilitation. Future work will involve clinical validation with post-stroke and cardiac patients.

## 1. Introduction

Physical therapy plays an important role in medical rehabilitation and is an integral part of it [1]. It helps patients recover from various medical conditions by improving their physical function strength and mobility.

Many injuries or medical conditions can affect the ability to function [2], including

Brain disorders, such as stroke, multiple sclerosis, or cerebral palsy;Long-term (chronic) pain, including back and neck pain;Major bone or joint surgery;Severe arthritis becoming worse over time;Severe weakness after recovering from a serious illness (such as infection, heart failure, or respiratory failure);Spinal cord injury or brain injury.

Physical rehabilitation describes the process that a person goes through to reach optimal physical functioning.

Controlling the rehabilitation process involves regular monitoring and adjustments based on patient progress. This ensures that the rehabilitation program is tailored to the individual’s needs, promoting optimal recovery.

Physiotherapists often face various challenges such as resource limitations, complex patient needs, communication barriers, financial constraints, and ethical and legal issues.

Computer vision and artificial intelligence (AI) are revolutionizing the field of medical rehabilitation, particularly in the control and monitoring of exercise. These technologies offer several advantages that enhance the effectiveness and efficiency of rehabilitation programs. Computer vision systems can be used to monitor patients’ movements during exercise therapy sessions. Unlike wearable or directly printed on garments sensors [3], video cameras do not restrict movement and do not require lengthy preparation for a rehabilitation session. By using cameras and depth sensors, these systems can track the movements of patients, ensuring that exercises are performed correctly and safely. This real-time feedback helps in

Correcting Form: Ensuring that patients perform exercises with the correct form to prevent injuries and maximize the benefits of the therapy;Progress Tracking: Monitoring patient improvements in range of motion, strength, and endurance over time;Remote Supervision: Allowing therapists to supervise patients remotely, which is particularly useful for those who cannot frequently visit rehabilitation centers;Reduce the workload of the physiotherapist: Allowing them to manage more patients simultaneously and focus on complex cases.

Although numerous patient-centered systems have been developed for home rehabilitation, there is a notable lack of systems designed to support both the physiotherapist and the patient [4].

In modern physical rehabilitation protocols, patients typically perform exercises with periodic feedback or guidance following initial demonstrations by the physiotherapist. As the patient tries to repeat the exercises after the physiotherapist, how accurately they can do this can serve as an important indicator of the degree of their rehabilitation.

The main objective of this work is to build a system of interactions between a patient and a physiotherapist using a quantitative assessment of the degree of similarity between their performance of exercises based on computer vision methods.

The nature of the exercises allows for the phases of movements of the therapist and the patient to be aligned in order to assess the dissimilarity of poses in the corresponding frames. We use a skeletal description of a human pose to form the basis for measuring such value. In [5], we have proposed the measure of dissimilarity between two skeletons, which can be used here for dual purposes. First, it can serve for time warping to align the frames in video sequences of the therapist and patient, which correspond to the same phases of an exercise. Second, it can measure the degree of dissimilarity between poses in aligned frames. Thus, we have two parts in the overall dissimilarity measure between the videos. One part is responsible for inconsistency in the dynamics of an exercise and another reflects the differences in posture. The final measure is a kind of balance between these two parts. Given the adopted protocol, it is essential not to interfere with the patient’s exercise execution during recording, as any real-time intervention could bias the evaluation. Therefore, the unit of assessment here is the complete the exercise recording and evaluate it against the instructor’s reference performance.

The main contributions of the proposed method can be summarized as follows.

We do not use any pre-alignment or registration techniques before computing the similarity between the two video sequences;We introduce a novel algorithm called Human Skeleton-based Balanced Time Warping, a modified dynamic time warping (DTW) algorithm-based technique for comparing two video sequences. Using skeleton-based dissimilarity allows us to significantly speed up and reduce the requirements for the computing resources of the warping algorithm;Our proposed method demonstrated the possibility of accurately measuring the rehabilitation progress of patients by quantitatively assessing how well patients reproduce the exercises performed by instructors;We collected a new dataset with objective scores of rehabilitation degrees.

The core innovation lies in the fact that, instead of using specific pose features, we adopted a featureless approach, elaborated in our previous works on skeleton standardization [6], fall detection [5], and skeleton distance evaluation [7], based on a balanced dissimilarity measure designed explicitly for skeletal model analysis in rehabilitation systems.

We hope that the integration of computer vision and AI technologies in exercise therapy represents a significant advancement in the field of medical rehabilitation. These technologies not only enhance the precision and effectiveness of rehabilitation programs but also make them more accessible and personalized. As a result, patients can achieve better outcomes and a higher quality of life.

The rest of the paper has the following structure. A brief review of the recent methods of human action analysis based on skeleton descriptions and dynamic time warping is presented in Section 2. Section 3 provides a comprehensive description of our methodology. The experimental evaluation and proof of the method are presented in Section 4.

## 2. Literature Review

Recent advances in human action recognition (HAR) have increasingly leveraged skeletal data due to its robustness to appearance variations and environmental conditions. In this section, we present a brief literature review of the recent works on HAR to clarify methodological trends, distinguish between general HAR and clinical applications, and highlight the evolution from handcrafted features to self-supervised deep models.

Early approaches to skeleton-based HAR relied heavily on handcrafted features combined with dynamic time warping (DTW) or related alignment strategies to handle temporal variability in motion sequences. Tormene et al. [3]. introduced a DTW variant for matching incomplete time series in post-stroke rehabilitation, establishing a foundational approach for variable-length motion comparison.

Kumar et al. [8] extracted skeletal joint trajectories from Kinect v2 data and applied graph-based time-series matching to recognize actions. Similarly, Vishwakarma and Jain [9] projected 3D joint positions onto a 2D plane to form movement polygons, enabling geometric feature extraction; their method achieved up to 95.7% accuracy across four general HAR benchmark datasets (MSR Action3D, Berkeley MHAD, TST Fall Detection, NTU-RGB+D).

İnce et al. [10] introduced wavelet transform (HWT), which is used to preserve the information of the features before reducing the data dimension. Dimension reduction using an averaging algorithm is also applied to decrease the computational cost, which provides a faster performance. Before the classification, the authors proposed a thresholding method to extract the final feature set. Finally, the K-nearest neighbor (k-NN) algorithm is used to recognize the activity with respect to the given data. In this case, the accuracy of the proposed model with LSTM was 82.2%.

Muralikrishna et al. [11] based his proposed work on human action recognition by combining structural and temporal features. He based his research on four datasets, namely, KTH, UTKinect, and MSR Action3D datasets. The overall accuracy of this method is 90.33%.

Ahad et al. [12] proposed the Linear Joint Position Feature (LJPF) and Angular Joint Position Feature (AJPF) based on 3D linear joint positions and angles between bone segments, which were then combined into two kinematics features for each video frame. These features include the variation in motion in the temporal domain, as each body joint represents kinematic positions and orientation sensors. Then, the extracted KPF feature descriptor was extracted using a low-pass filter, which consists of temporal domain statistical features, which were further segmented using Position-based Statistical Features (PSFs). For classification purposes, a variety of classifiers including Support Vector Machine (Linear), RNN, CNNRNN, and ConvRNN models were used. The highest classification rate was 98.44% using ConvRNN + PSF.

The above works were based mainly on the extraction of features from human action datasets and classification using deep learning models. The exact application of the above methods in the field of biomedical engineering is still to be proposed. More recent works exploit deep architectures to learn spatiotemporal representations directly from raw or minimally processed skeleton sequences.

Wang et al. [13] proposed an algorithm “MEET_JEANIE” where the 3D skeleton sequences whose camera and subjects’ poses can be easily manipulated were extracted and evaluated on skeletal Few-shot Action Recognition (FSAR), where matching the temporal blocks of support–query sequence pairs (by factoring out nuisance variations) is essential due to the limited samples. Given a query sequence, the author’s created several views by simulating several camera locations. For comparison purposes, the smallest distances amongst matching pairs were selected, with different temporal viewpoint warping patterns. The algorithm achieved 65% accuracy over the already existing ones. In the above work by Wang et al. [13], an alignment procedure was also used.

Graph Convolutional Networks (GCNs) have emerged as a powerful tool for modeling skeletal topology. Du et al. [14] had proposed a self-supervised GCN framework for rehabilitation exercise assessment, leveraging regularization to improve generalization without extensive, labeled data. The experiments on the existing HAR benchmark dataset validated that the proposed methods achieved state-of-the-art performance with lower prediction and improved performances.

Park et al. [15] generated motion embedding vectors for the body parts, and the motion variation loss was introduced in order to distinguish similar kinds of motions. The authors also worked on the synthetic dataset to train the model. The entire system was tested on the NTU RGB+D 120 dataset.

Chen et al. [16] target fine-grained activity recognition and prediction in smart manufacturing, aiming to improve productivity and safety. It introduces the two-stage deep learning network combining multi-modal feature extraction (RGB and hand skeleton) with temporal modeling (LSTM and GRU), achieving high accuracy in both trimmed and continuous assembly activity videos.

While many HAR methods focus on generic action classification, a growing body of work tailors techniques to clinical rehabilitation, where precision, interpretability, and patient-specific adaptation are critical.

Anton et al. [17] used “KiReS”, a telerehabilitation system based on Kinect, using 3D models to check how an exercise should be repeated by patients, as performed by the instructor. This feature can help them improve the performance of the exercises performed.

Lam et al. [4] implemented an Automated Rehabilitation System (ARS) in a hip/knee replacement clinic, providing real-time visual feedback by superimposing ideal motion trajectories onto patient performance.

Çubukçu et al. [18] proposed a Kinect-based integrated physiotherapy for shoulder damage. The authors implemented the method on 14 volunteers who were treated with conventional methods and 15 who were treated with the proposed mechanism. Yurtman et al. [19] proposed an automatic system to detect and evaluate physical therapy exercises. The accuracy is 93.46% for exercise classification and 88.65% for simultaneous exercise.

Adolf et al. [20] assessed the quality of the Single Leg Squat Test and Step-Down functional tests. The authors recorded the exercises using forty-six healthy subjects, extracting the motion data, and classifying them for assessment by three independent physiotherapists. The authors calculated the ranges of movement in key point-pair orientations, joint angles, and the relative distances of the monitored segments using machine learning. The results showed that the AdaBoost classifier achieved a specificity of 0.8, a sensitivity of 0.68, and an accuracy of 0.7.

Gal et al. [21] presented a Kinect and fuzzy inference system-based e-rehabilitation system. The Kinect could detect the motion of patients, and the fuzzy inference system could interpret the data by looking through the initial posture and motion changes on a cognitive level. The system is capable of assessing the initial posture and motion ranges of 20 joints. Using angles to describe the motion of the joints, exercise patterns were developed for each patient.

Notably, while the entertainment industry has long known solutions for the Xbox that allow one to assess the degree of similarity of the repetition of dance movements for a typical rhythm game avatar (Just Dance, Dance Central), rehabilitation requires domain-aware similarity metrics that account for the initial posture, movement type, pace, and therapeutic intent—factors often ignored in general-purpose skeleton matching [22].

Liao et al. [23] conducted a study which reviews an important role performed by trained clinicians in assisting patients with performing rehabilitation tasks. Tsiouris et al. [24] conducted a review on virtual coaching designed to initiate healthcare interventions, which combined sensing and system-user interventions. The results focused on the fact that home coaching techniques were better to deal with healthier lifestyles and training patients with IOT devices and sensors. Debnath et al. [25] further underscore the need for clinically validated, sensor-driven coaching systems that close the feedback loop between patients and healthcare providers.

The exercises in physical rehabilitation have their own specific movement patterns, which include the initial positioning of the patient, the predominant types of movements used, and the pace of the exercises. The dissimilarity function proposed in this work, unlike general-purpose skeleton-based measures (e.g., [22]), allows us to take these characteristics into account to improve the accuracy and reliability of assessing the patients’ rehabilitation progress. The difference between the already existing methods and the present method is that, here, we have a specific goal that could be applied to patients who have suffered from motion disorders. For this reason, we have extracted the skeletons of the instructor and the patient and found the degree of similarity using a novel algorithm termed as the “Human Skeleton-based Balanced Time Warping algorithm”.

## 3. Proposed Methodology

We consider here the degree of rehabilitation of the patient based on his ability to perform certain physical exercises at the instructor’s command. The more accurately he repeats the instructor’s movements, the better the rehabilitation goes.

We will apply our algorithm to the frames of the two video sequences, obtain the alignment path between frames using the distance between skeletons, and, finally, compute the distance between video sequences with the help of the distances between skeletons for each corresponding pair of frames in the alignment path. The general architecture of the system is shown in Figure 1.

The first step of the proposed method is extraction of a skeleton from each frame of a video sequence. Following the convention in [6], all skeletons are normalized and transformed into a standard form. Using the instructor’s reference sequence, we compute per-coordinate attention weights that reflect the relative importance of each spatial dimension of each skeletal point for the specific exercise being performed. Concurrently, we quantify the intensity of motion over time for both the patient and the instructor by analyzing temporal derivatives of joint trajectories.

To account for differences in movement timing, we apply dynamic time warping (DTW) to align the two sequences, ensuring that frames corresponding to the same phase of the exercise are matched. This alignment enables the computation of a dynamics-based dissimilarity component, which captures discrepancies in movement tempo and rhythm.

Subsequently, for each pair of temporally aligned skeletons, we compute a weighted Euclidean distance [7] between the patient’s and instructor’s poses. The weights are derived from the previously estimated attention scores, emphasizing clinically or kinematically relevant joints and dimensions. This yields a pose-based dissimilarity component that reflects deviations in spatial configuration.

The total dissimilarity between the patient and instructor video sequences is defined as a balanced sum of the dynamics-based and pose-based components, providing a comprehensive measure of exercise fidelity that accounts for both temporal execution and postural accuracy.

A more comprehensive and detailed description of the proposed method is provided below.

The proposed method begins by extracting a skeletal representation—referred to as a skeleton—from each frame of a video sequence. Skeleton models are widely used in various human activity analyses and recognition systems, like fall detection, activity classification, human–computer interaction, etc. There are several modern techniques to obtain a skeletal model from a video frame. First, depth sensors, as they became more available, occupy an increasing place in event recognition systems. Many of them can directly produce a skeletal description of the human figure for compact representation of a person’s posture. An alternative way consists of using special neural network-based solutions that build skeleton models from images produced by a conventional RGB camera like YOLO11-pose, Alphapose, Google MediaPipe Pose Landmarker, and others. All these methods produce a skeleton model in different forms (Figure 2).

The method of constructing a skeletal model from a video frame is not the subject of this work. Considering our previous experience, we adopt here the standardized skeleton models obtained by the special procedure described in [6]. Each skeleton model is actually a graph and can be described by a correspondent adjacency matrix. The problem of converting one skeleton model to another could be formally stated as a problem of adjacency matrix transformation. Paper [6] divides all forms of skeletal models into three categories and provides a number of simple equations for each category for conversion. Such a procedure allows them to transform different skeletal models into the standard form (Figure 2), making the proposed degree of rehabilitation assessment independent from the type of sensor. After that, each skeleton comprises 17 vertices (Figure 2), corresponding approximately to major body joints, with each point represented by its 3D coordinates.

In particular, the experimental dataset of videos corresponding to the instructor and the patients who try to successfully execute the exercises demonstrated by the instructors was collected using the Kinect v2 RGBD sensor (Microsoft Corporation, Redmond, WA, USA).

A dissimilarity measure between two skeletal models plays a key role in the development of any activity assessment system. In our case, the specificity of physical exercises for the rehabilitation programs allows us to improve the reliability and robustness of the measure considering the different contributions of the skeleton points to the final movement during a particular exercise, producing motion-dependent dissimilarity measures.

In paper [7], we proposed to use weighted Euclidean distance to calculate the dissimilarity measure, with weights proportional to the standard deviation of the coordinates of the correspondent skeletal point representing the degree of attention to the different coordinates of different skeleton points. The general flowchart of the dissimilarity measure evaluation can be found in Figure 3.

The distance between pairs of skeletons, I (instructor) and P (patient), could be found by the motion-dependent dissimilarity measure:$$d\left[S(I),S(P)\right]=\frac{1}{K}\sum_{p=0}^{K-1}\sqrt{\sum_{m\in \{x,y,z\}}w_{p}^{m}{\left(S{\left(I\right)}_{p}^{m}-S{\left(P\right)}_{p}^{m}\right)}^{2}},\;w_{p}^{m}=\frac{\sigma_{p}^{m}}{\sum_{m\in \{x,y,z\}}\sum_{p=0}^{K-1}\sigma_{p}^{m}},$$
where K—number of used points of skeletons, $S{\left(I\right)}_{p}^{m}$—m-th coordinate of p-th point of instructor skeletal model S(I), $S{\left(P\right)}_{p}^{m}$—m-th coordinate of p-th point of patient skeletal model S(P), $w_{p}^{m}$—weights of importance, representing the degree of attention to the m-th coordinate of the p-th point, and $\sigma_{p}^{m}$—standard deviation of m-th coordinate of the p-th point of instructor skeleton.

Our challenge is to measure the distance between two video sequences based on the distance between skeletons. But video sequences have different lengths, and the relative time of the beginning of an exercise is often not well-defined. The common solution is to use a version of dynamic time warping (DTW) to find a match between frames of the first video sequence and frames of the second video sequence to obtain an alignment path between them. The comprehensive review of different approaches to temporal alignment methods can be found in [3,23].

The problem with aligning instructors’ and patients’ records can be approached from two opposing perspectives. On the one hand, a skeletal model with 17 points in three-dimensional space yields 51 curves representing changes in corresponding coordinates over time, and multivariate temporal alignment could be performed by comparing these curves to an equal number of curves generated by the second record being compared. On the other hand, for alignment, it is sufficient to use a distance function between states of the skeletal models at comparable moments in time without directly using changes in the coordinates of the skeletal model points over time. We propose the use of the second scheme for temporal alignment. This makes the algorithm independent of how skeletal models are compared and lets us use motion-dependent dissimilarity measures for both sequence alignment and calculating the person-pose discrepancy in frames that match as a result of alignment.

The classical DTW algorithm cannot handle the time shift between the start of exercises on the compared records, as well as the different end times of the exercises. Although there exist open-ending and open-beginning versions of DTW [3], experimental evaluation shows the lack of stability of these methods while using dissimilarity measures between skeletal models for matching. We attempt to circumvent this issue by formulating the following statement of the temporal alignment task. As a common practice, we will denote one sequence $R^{b}=(R_{i}^{b},i=0,\ldots ,N^{b}-1)$, specifically the shortest one with length $N^{b}$, as the base and the other $R^{r}=(R_{i}^{r},i=0,\ldots ,N^{r}-1)$ as the reference video sequence with length $N^{r}$. As a result, at each frame i in the base sequence, there should be a determined value $v_{i}$ indicating the time shift between corresponding points in the base i and reference $i+v_{i}$ sequences. The result of the analysis is represented by the set $V=\left(v_{i},\;i=0,\ldots ,N^{b}-1\right)$, which takes values from the finite number of possible shifts $v\in \left\{-w/2,-w/2+1,\ldots ,w/2\right\}$, where w defines the admissible shifts range. Parameter w sets the size of the shift window and specifies a possible delay between the start time of the exercises by the instructor and the patient, as well as between the end time of the exercises.

As well as in the DTW algorithm, we use the optimization approach to the alignment task with the following loss function:(1)$$J\left(V\left|R^{b},R^{r}\right.\right)=\sum_{i=0}^{N^{b}-1}d\left[S(R_{i}^{b}),S(R_{i+v_{i}}^{r})\right]+\sum_{i=1}^{N^{b}-1}\gamma \left(v_{i-1},v_{i}\right).$$

The loss function $J\left(V\left|R^{b},R^{r}\right.\right)$ consists of two parts, where $d\left[S(R_{i}^{b}),S(R_{i}^{r})\right]$ is the distance between two skeletons: $S(R_{i}^{b})$ on frame number $R_{i}^{b}$ of the medical instructor record and $S(R_{i}^{r})$ on frame number $R_{i}^{r}$ of the patient record, while $\gamma \left(v_{i-1},v_{i}\right)$ represents the smoothness term, which reflects that sequential interframe displacements have to change smoothly over time, and no displacement crossing is allowed based on the physical nature of the process. Here, we take $\gamma \left(v_{i-1},v_{i}\right)$ in the widely used form of a quadratic function with parameter q if the transition between displacements is valid and there are no cross references or it provides a big penalty overwise:$$\gamma (v_{i-1},v_{i})=\left\{\begin{array}{c} q\cdot {\left(v_{i-1},v_{i}\right)}^{2},\;v_{i}\geq v_{i-1}-1,\\ M,v_{i}<v_{i-1}-1, \end{array}\right.$$
where q is a smoothness penalty; M is a sufficiently large constant.

Such kinds of smoothness terms prevent changes in the order of frames in an aligned sequence but also penalize abrupt changes in interframe shifts.

Criterion (1) belongs to the class of pairwise separable functions, since it is a sum of functions of no more than two arguments. A problem of such kind is well-known in computer vision as the graph-based energy minimization problem [26]. Here, we have a rather simple case of a process in discrete time, and criterion (1) corresponds to a chain-like graph of variable adjacency. To solve it, a dynamic programming procedure [27] can be used, which has a linear computational complexity with respect to the length of the base sequence.$$\hat{V}=\underset{V}{\arg\min}J\left(V\left|R^{b},R^{r}\right.\right).$$

Having a set of shifts $V=\left(v_{i},\;i=0,\ldots ,N^{b}-1\right)$, it is easy to obtain a sequence of pairs of corresponding frames $R=(R_{0}^{b},R_{0}^{r}),\ldots ,(R_{N^{b}-1}^{b},R_{N^{b}-1}^{r})$, where $\left(R_{i}^{b},R_{i}^{r}\right)$ represent the indexes of matched frames for reference number i, b means that the frame is from the base video with length $N^{b}$, r means that the frame is from a reference video with length $N^{r}$, and N is the total number of references in the alignment path.

Considering that the shortest video is taken as «base», the total distance between two videos should be calculated as(2)$$D\left(R^{b},R^{r}\right)=\frac{1}{N^{b}}\left\{\sum_{i=0}^{N-1}d\left[S(R_{i}^{b}),S(R_{i}^{r})\right]+\alpha \sum_{i=1}^{N-1}\left|\left(\left|R_{i}^{b}-R_{i}^{r}\right|\right)-\left(\left|R_{i-1}^{b}-R_{i-1}^{r}\right|\right)\right|\right\},$$
where $d\left[S(R_{i}^{b}),S(R_{i}^{r})\right]$ is the distance between two skeletons: $S(R_{i}^{b})$ on frame number $R_{i}^{b}$ of the medical instructor record and $S(R_{i}^{r})$ on frame number $R_{i}^{r}$ of the patient record.

The dissimilarity between two skeleton models on two frames is determined by the differences in poses but also by the discrepancy between the anthropometric characteristics of the instructor and the patient. We assume that the minimal dissimilarity value among all frame pairs $M=\min d\left[S(R_{i}^{b}),S(R_{i}^{r})\right],\;i=0,\ldots ,N-1$, considering preliminary normalization of the skeletons, is just determined by the anthropometric discrepancy. Therefore, the final distance should be corrected.

So, the total distance between two videos in a whole should be calculated as(3)$$D\left(R^{b},R^{r}\right)=\frac{1}{N^{b}}\sum_{i=0}^{N-1}\left\{d\left[S(R_{i}^{b}),S(R_{i}^{r})\right]-M\right\}+\alpha \frac{1}{N^{b}}\sum_{i=1}^{N-1}\left|\left(\left|R_{i}^{b}-R_{i}^{r}\right|\right)-\left(\left|R_{i-1}^{b}-R_{i-1}^{r}\right|\right)\right|.$$

The first term in the criterion $DP\left(R^{b},R^{r}\right)=\frac{1}{N^{b}}\sum_{i=0}^{N-1}\left\{d\left[S(R_{i}^{b}),S(R_{i}^{r})\right]-M\right\}$ is responsible for the difference between the instructor’s and the patient’s performance in the form of posture, the second term $DT\left(R^{b},R^{r}\right)=\frac{1}{N^{b}}\sum_{i=1}^{N-1}\left|\left(\left|R_{i}^{b}-R_{i}^{r}\right|\right)-\left(\left|R_{i-1}^{b}-R_{i-1}^{r}\right|\right)\right|$ evaluates the discrepancy in the dynamics of performance over time, and α is a balancing coefficient.

While measure $D\left(R^{b},R^{r}\right)$ represents the dissimilarity between different records, the intensity of movements during the exercise can be an important characteristic of a single record in the context of rehabilitation. Based on this observation, the intensity values can be used as a standalone characteristic, as well as contribute to the dynamic part $DT\left(R^{b},R^{r}\right)$ in the criterion (3). In the latter case, we should use the inverse values of intensities:(4)$$DT\left(R^{b},R^{r}\right)=\frac{1}{N^{b}}\sum_{i=1}^{N-1}\left|\left(\left|R_{i}^{b}-R_{i}^{r}\right|\right)-\left(\left|R_{i-1}^{b}-R_{i-1}^{r}\right|\right)\right|+\beta (E_{\max}-E^{r}),$$
where $E^{r}$ is an intensity of the patient’s record; $E_{\max}$ is the maximum intensity in a dataset.

We use an average interframe dissimilarity of a record as an estimation of intensity:(5)$$E^{r}=\frac{1}{NK}\sum_{i=1}^{N-1}\sum_{p=0}^{K-1}\sqrt{\sum_{m\in \{x,y,z\}}{\left(S{\left(R_{i}^{r}\right)}_{p}^{m}-S{\left(R_{i-1}^{r}\right)}_{p}^{m}\right)}^{2}}.$$Therefore, the final metric between instructor and patient records has the form of a linear combination of two components with the coefficient α:(6)$$D\left(R^{b},R^{r}\right)=DP\left(R^{b},R^{r}\right)+\alpha \cdot DT\left(R^{b},R^{r}\right).$$

We will pay attention to the selection of the value of the α parameter in Section 4.3.

## 4. Experimental Study and Results

### 4.1. Dataset

The primary target of this study is to provide an objective and reliable measure of a patient’s rehabilitation progress by comparing their exercise performance against a reference recording performed by an instructor. This goal defines the basic requirements for the dataset for experiments.

Existing open-access datasets, such as FineRehab by Li et al. [28] (50 participants, 16 rehabilitation actions), the UCO Physical Rehabilitation Dataset by Aguilar-Ortega et al. [29] (27 participants, eight exercises), IntelliRehabDS by Miron et al. [30] (29 participants, nine rehabilitation gestures), and the University of Liverpool Rehabilitation Exercise Dataset (UL-RED) by Reji et al. [31] (10 participants, 22 actions), primarily contain motion recordings of both healthy individuals and patients. However, these datasets lack rehabilitation scores and thus effectively support only a binary distinction (e.g., healthy vs. impaired), offering limited granularity in assessing rehabilitation quality. In contrast, the KIMORE dataset by Capecci et al. [32] (78 participants, five rehabilitation exercises) stands out by providing expert-rated clinical scores derived from a standardized questionnaire, as well as reference recordings of expert (therapist) performances. While KIMORE is arguably the most suitable dataset for rehabilitation assessment tasks, its clinical ratings, though standardized, remain inherently subjective due to inter-rater variability.

In this work, rather than estimating a rehabilitation score from a patient’s recording, we adopt a different approach: we aim to simulate exercise execution at varying levels of motor impairment. To this end, we introduce a new dataset specifically designed to evaluate the proposed dissimilarity measure. Moreover, the collection of the dataset let us create, debug, and test the necessary components of the real-world automated exercise control system.

Our dataset comprises video recordings of a physiotherapist (instructor) and 22 participants (17 men and 5 women, aged 22–55 years) in a controlled laboratory environment. In our approach, the instructor’s recording was presented to the patient in real time during the exercise execution. Each person repeated the instructor’s exercises three times. The first time, they did it as accurately as possible. The second time, they did it less precisely, simulating a patient with minor motor impairments. The third time, they did it even less accurately, simulating a patient with severe motor impairments. We will denote these three accuracy levels of repetition as “good”, “intermediate”, and “bad”. Thus, the total number of records in the dataset (including the instructor) is equal to 136.

The main limitation of such a protocol of three-level estimations of performance accuracy, while improving reliability, turns the problem into a problem of ranking regression, as the ground truth is represented on the ordinal scale.

All data were captured using the Microsoft Kinect v2 sensor and include 3D coordinates of 17 skeletal joints (corresponding to the standard Kinect v2 skeletal model) for every video frame (see Figure 2).

### 4.2. Evaluation of the Relationship Between the Level of Accuracy of Exercise Performance and the Intensity of Movements

To evaluate the relationship between the level of accuracy of the exercise performance and the intensity of movements, we calculate the intensity of each record in the dataset by Equation (5). Figure 4 shows the strong relations between the «quality» of exercise (good, intermediate, and bad) in mimicking the instructor’s patterns and the estimation of the intensity of exercises.

### 4.3. Video Sequence Distances Visualization and Analysis

In this section, we enlist the experimental results as to how efficiently the patients repeat the exercises performed by the instructors.

We deliberately decompose the total distance between two videos (3) into two components: a pose dissimilarity measure DP and a dynamics dissimilarity DT. This approach allows us to avoid difficulties associated with visualizing isolated data points of metric similarity to the reference and instead allows for the effective visualization and analysis of the experimental results on the collected dataset in a two-dimensional representation. This enhances the interpretability of the patient’s rehabilitation progress, facilitates the construction of individual rehabilitation trajectories, and enables comparison across patients within the dataset.

In the figures, the second characteristic (DT) is plotted along the ordinate axis and multiplied by 10 (see Figure 5). It is clear that the closer the patient’s point is to the instructor’s point, the better the patient performs the exercises. It should be noted that individual exercises performed in both sitting and standing positions formally generate two isolated dissimilarity spaces [33].

Decomposing the function D into two components, DP and DT, provides a clearly interpretable and intuitive visualization of each exercise act on a two-dimensional plane. Obviously, the instructor will be displayed at the origin. However, it should be understood that this will result in a separate visualization for each exercise. However, we wanted exercises of different types (in our case, sitting and standing positions) to be projected onto a single display plane. To this end, we performed separate centering and scaling of the points for each exercise. In this case, we obtained projections of the points onto a single visual display field simply by placing the individual centers of each exercise at the origin. We were somewhat pleasantly surprised that no additional alignment of the circles (sitting position) and triangles (standing position) was required.

Connected lines correspond to the same specific patient identified as P_18.

The reference position of the instructor’s exercises is marked in purple in Figure 5. After some time, the instructor repeated the exercises, attempting to perform the movements as accurately as possible in comparison to the first recording. This repetition is represented in the figure by objects with a purple outline. Simulations of “intermediate” and “bad” accuracy levels by the instructor were not conducted.

As can be seen from Figure 5, there is good clustering of points belonging to different accuracy levels (“good,” “intermediate,” and “bad”). We tried to highlight the areas of concentration of rehabilitation levels, and, to avoid drawing it manually, we used a classifier based on a simple neural network. We consider such division on the colorful areas to be useful for the therapist. Classification was carried out using the Multi-layer Perceptron Classifier from the Scikit-learn python library [34] with standard training parameters.

On one hand, in our collected dataset, we assess how closely a patient performs an exercise relative to the instructor using an ordinal scale ranging from “bad” to “intermediate” to “good.” On the other hand, when it comes to rehabilitation outcomes, progress can look very different from one patient to another, both in pace and pattern. As such, directly comparing numerical recovery scores across individuals—especially in terms of overall improvement or specific physiological functions—remains a complex challenge that warrants further investigation.

### 4.4. Performance Metrics and Hyperparameter Values Estimation

As stated in Section 4.1, each patient record in the experimental dataset is associated with a specific level of accuracy in reproducing the instructor’s movements, which naturally turns the problem into a ranking regression task. At the same time, the primary objective of this work is to produce a quantitative measure of the distance between the recorded executions of the therapeutic exercise by the instructor and the patient. This raises the issue of selecting an appropriate method for evaluating the performance of the proposed distance estimation approach.

Note that the dissimilarity (or distance) function, together with a fixed reference element that the instructor naturally acts as, induces a natural (non-strict) order on the set of distances, because this set is a subset of the real numbers, which are already totally ordered. This induces a pre-order on the set of patient records for a particular exercise:$$R_{i}^{r}\underline{≺}R_{j}^{r}\Leftrightarrow D\left(R^{b},R_{i}^{r}\right)\leq D\left(R^{b},R_{j}^{r}\right),$$
where $R_{i}^{r}$ and $R_{j}^{r}$ are records of the same exercise of the patient r.

In this work, we focused on preserving a meaningful, monotonic relationship between increasing the dissimilarity of movements and decreasing the performance quality. In other words, as the mismatch in pose and dynamics grows, the score should reliably reflect a drop in the accuracy level.

The commonly adopted way to capture this is to use Spearman’s rank correlation coefficient [35]:(7)$$r_{s}=1-\frac{6\sum_{i=1}^{n}d_{i}^{2}}{\;n\left(n^{2}-1\right)},$$
where n is the number of records for each person in each position in the dataset; d is the difference between the ranks of records. It is assumed that the “good” execution of an exercise corresponds to rank 0, an “intermediate” execution corresponds to rank 1, and a bad execution corresponds to rank 2. In our case, each person performs an exercise three times in the standing position and in the sitting position, n=3. Therefore, Equation (7) takes a simple form:(8)$$r_{s}=1-\frac{\sum_{i=1}^{3}d_{i}^{2}}{4}.$$

To obtain a measure of performance on the whole dataset, we average the correlation coefficient (8) over records of all people in both positions.

Such a measure lets us choose the optimal parameter α in (6) to coordinate the importance of pose (DP) and dynamic (DT) dissimilarity. Considering the way we adopt visualization, it will be more convenient to represent Equation (6) in a slightly different form:(9)D=DP⋅cosφ+DT⋅sinφ.

Equation (9) defines the projection of each record onto the axis passing through the origin at an angle φ to the horizontal. Figure 6 demonstrates the dependency between the projection angle φ and the average Spearman’s rank correlation coefficient.

The maximum average Spearman’s rank correlation coefficient is equal to 0.977 and is achieved for a range of φ angles; for certainty, we chose the smallest of them with the angle value $\varphi =13^{{}^{\circ}}$. The correspondent axis is shown in Figure 5 as a blue line. And the distribution of the number of records of each type on that axis is represented in Figure 7.

Results of the experimental evaluation of the proposed measure are shown in Figure 8. Each line corresponds to the records of certain patients (P_patient number) in the dataset, suffix _01 corresponds to the standing position, and _02 corresponds to the sitting position. The color markers (dots for sitting, triangles for standing exercises) reflect the exercise execution levels: the green color for “good”, blue color for “intermediate”, and red color for “bad”. The proper sequence should be green, blue, and red. A different order of colors indicates an error. Distances between markers reflect the distance D to the record of instructor execution, assigning zero to the minimal distance.

It can be seen that there are only two records with the wrong order of execution quality, P_06_02 and P_18_02 (reminder: suffix _02 means sitting position), but in both cases, the distances between disordered states are very small.

To evaluate the sensitivity of the optimal angle to variations in the composition of the patient group, we employed an 11-fold cross-validation procedure. The entire dataset, excluding the instructor’s records, was randomly partitioned into 11 subsets, each containing all exercise recordings from two distinct participants. In each iteration of the experiment, one subset (12 records) was held out, and the optimal angle φ was selected via exhaustive search over the range in 1° increments using the remaining ten subsets (120 records). Figure 9 presents the plots of the mean Spearman’s rank correlation coefficient as a function of the angle obtained for all cross-validation folds.

The data in Table 1 demonstrates sufficient robustness of the estimated angle φ to variations in the patient group.

The angle φ yielding the maximum average Spearman’s rank correlation coefficient differed from the previously identified optimal value ($\varphi =13^{{}^{\circ}}$) in only two out of the eleven cross-validation folds. Even in those cases, the average Spearman’s rank correlation coefficient (see Figure 9) remains very close to the fold-specific optimum, indicating minimal practical impact on performance.

## 5. Conclusions and Discussion

This study presents a novel, vision-based system for the automated assessment of physical therapy exercise performance, grounded in a Human Skeleton-based Balanced Time Warping (HS-BTW) algorithm. Unlike conventional approaches that rely on predefined activity classes, deep learning models requiring large, annotated datasets or wearable sensors demanding calibration; our method offers a calibration-free, interpretable, and computationally efficient solution using only RGB-D or even RGB video input.

In pursuit of maximizing objectivity, our newly collected dataset eschews reliance on subjective therapist assessments or predefined feature sets, which are commonly employed in other datasets such as KIMORE [32]. Rather than relying on these established methods, we tasked actors with replicating instructional movements with different levels of accuracy, simulating varying degrees of motor impairments. Consequently, this unique form of data collection necessitates the utilization of specialized metrics tailored specifically to capture the ranked characteristics inherent within the evaluative framework. However, this innovative strategy simultaneously excludes compatibility with conventional benchmark datasets.

The core innovation lies in the fact that, instead of using specific pose features, we adopted a featureless approach [36,37] based on a balanced dissimilarity measure designed explicitly for skeletal model analysis. This measure separately quantifies posture deviation (DP) and dynamic inconsistency (DT), then combines them into a single metric optimized via Spearman’s rank correlation to align with rehabilitation quality. Note that DP uses the attention mechanism to the coordinates of points of the skeletal model. Its dual application includes both the temporal alignment of recorded videos against their associated skeletal representations and the direct quantification of deviations between corresponding poses across individual frames.

Evaluated on our dataset of 134 recordings from 22 participants simulating three levels of motor impairment (“good,” “intermediate,” and “bad”), the system achieved a remarkably high correlation of 0.977 between computed dissimilarity and execution quality. Visualization in the DP–DT space revealed clear clustering by accuracy level, demonstrating the method’s sensitivity to clinically meaningful differences in movement fidelity.

Compared to existing rehabilitation assessment systems—many of which depend on subjective clinician scores, fixed exercise templates, or sensor-laden setups—our approach provides objective feedback without requiring pre-alignment, extensive training data, or patient-specific calibration. By decoupling static posture errors from temporal dynamics, it also offers actionable diagnostic insights, enabling therapists to identify whether a patient struggles with form, timing, or both.

While some recent works like [16,20] and some others and the present work combine temporal and dynamics features when making decisions, our work addresses physical therapy monitoring, proposing a non-deep learning approach that uses RGB-D video and a customized balanced time warping algorithm to assess exercise quality. It focuses on comparing the motions of the patient and therapist using 3D skeletons. Here are some additional advantages of our method:(1)No Need for Extensive Training Data: Our work uses a similarity-based evaluation approach rather than supervised deep learning, reducing the need for large, annotated datasets, which are often expensive and time-consuming to collect.(2)Sensor-Free and Calibration-Free: Our work relies on RGB-D video and skeletal data without requiring wearable sensors or device-specific calibration, making it easier and more practical to deploy in clinical or home settings.(3)Quantitative and Interpretable Feedback: Our present method produces interpretable metrics (posture and dynamics scores) that directly reflect the quality of movements, providing meaningful, actionable feedback to patients and therapists.(4)Clarity in Error Assessment: By separating static posture (DP) and dynamic movement (DT) errors, the system helps identify specific areas where the patient deviates from the correct performance, enabling targeted rehabilitation.(5)Robust to Individual Variations: The algorithm measures relative similarity to a reference rather than forcing classification into fixed labels, making it more adaptable to individual differences in patient performance.(6)Supports Remote Monitoring: Non-reliance on heavy computation (e.g., deep neural networks) of our proposed work makes it suitable for tele-rehabilitation, allowing patients to receive feedback outside of clinical environments.(7)No Predefined Activity Classes Required: Our present proposed work depends on predefined activity labels for classification, thereby evaluating the quality of performance irrespective of activity type, allowing for more flexibility in usage.

These attributes make the proposed system particularly well-suited for tele-rehabilitation and home-based care, where accessibility, scalability, and interpretability are critical. Future work will focus on clinical validation with real post-stroke and cardiac patients, integration into therapist dashboards, and extension to a broader repertoire of therapeutic exercises.

On the whole, we understand that, although our dataset was collected under supervision and with the involvement of medical professionals, it cannot be used for clinical purposes without recordings from actual patients. We currently have an agreement in place with the University Medical Center and the City Clinical Hospital to expand our dataset and evaluate the system on real patients.

## Figures and Tables

**Figure 1 sensors-25-06696-f001:**
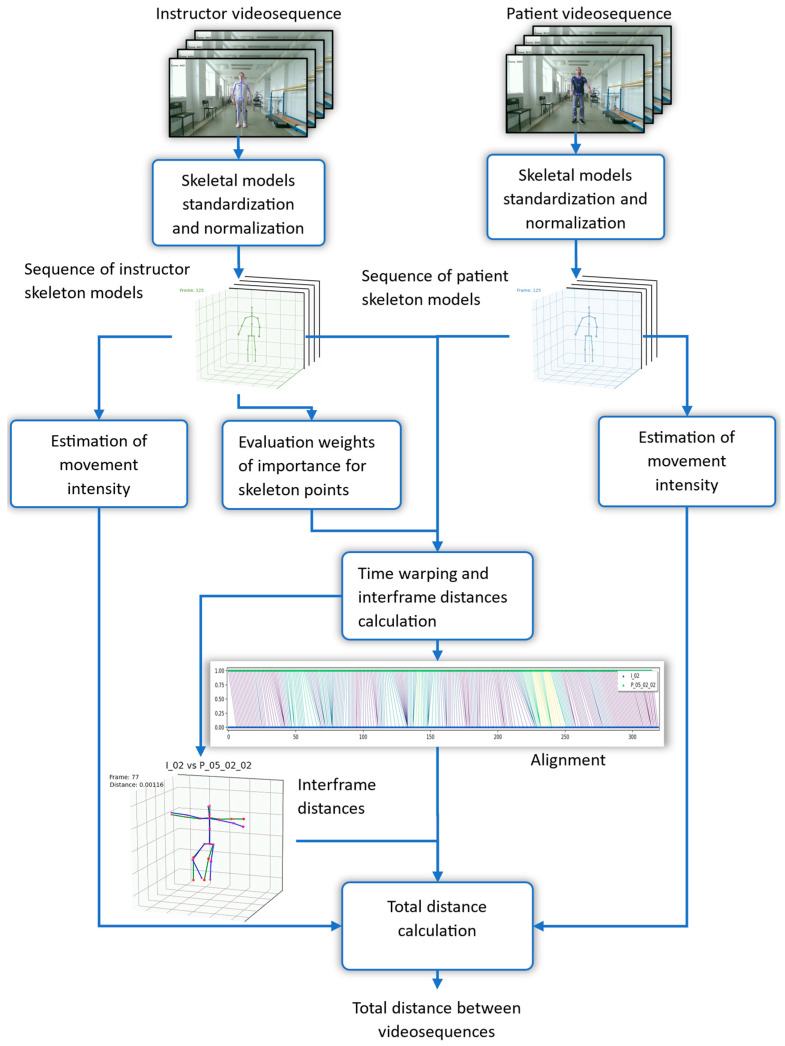
Flowchart of total distance between video sequences evaluation.

**Figure 2 sensors-25-06696-f002:**
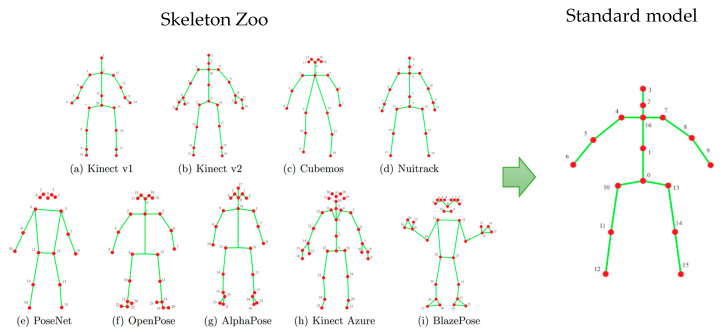
Standardization of the skeletal model.

**Figure 3 sensors-25-06696-f003:**
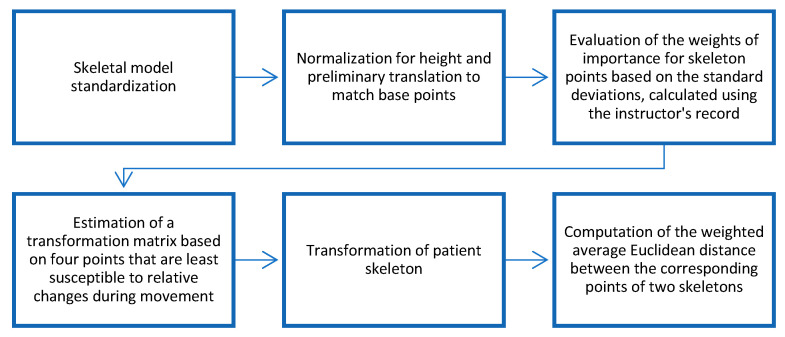
Flowchart of evaluation of the motion-dependent dissimilarity measure [7] between the two skeleton models.

**Figure 4 sensors-25-06696-f004:**
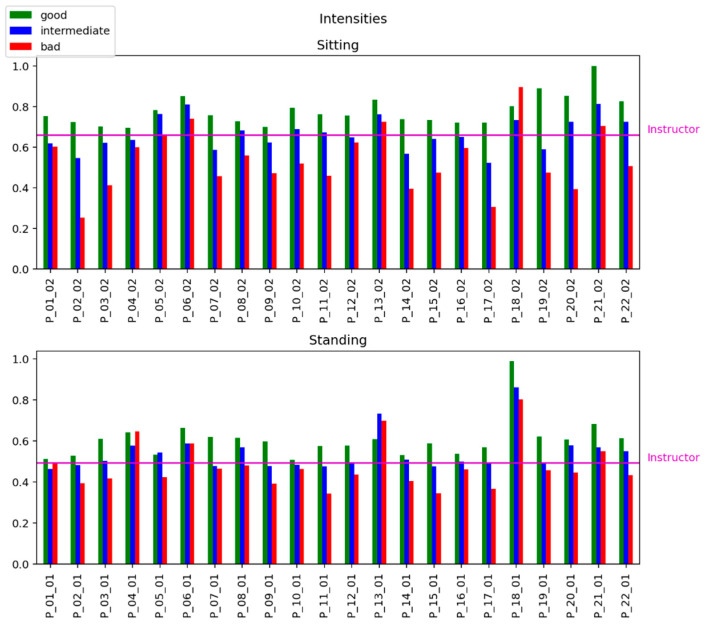
Normalized intensities of movements for 22 patients in sitting (top sub-picture) and standing (bottom sub-picture) positions under varying performance accuracy levels (“good,” “intermediate,” and “bad”).

**Figure 5 sensors-25-06696-f005:**
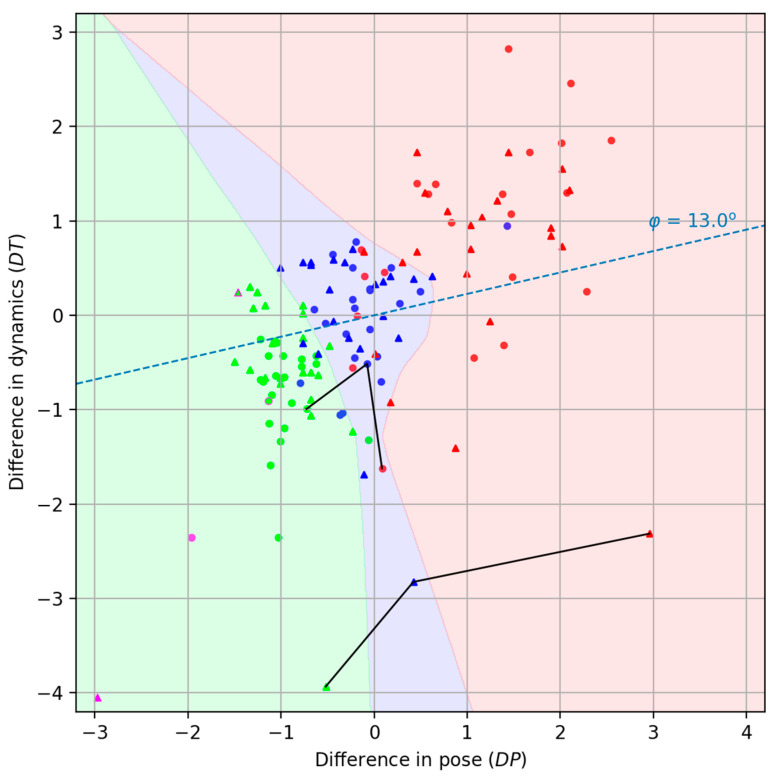
Presentation of data as normalized distances of individual patient exercises from the instructor’s reference exercises. The ordinate axis shows the *DT* value; the abscissa axis shows the *DP* value. Circles correspond to exercises in a sitting position, triangles to a standing position. Connected lines correspond to specific patient (P_18). Patients in sitting exercises are dots and in standing exercises are triangles. Green marks are the instructor repeating the exercise and 22 patients who have repeated the exercises well. Blue marks are intermediate patients. And finally, the red ones are the most inaccurate patients.

**Figure 6 sensors-25-06696-f006:**
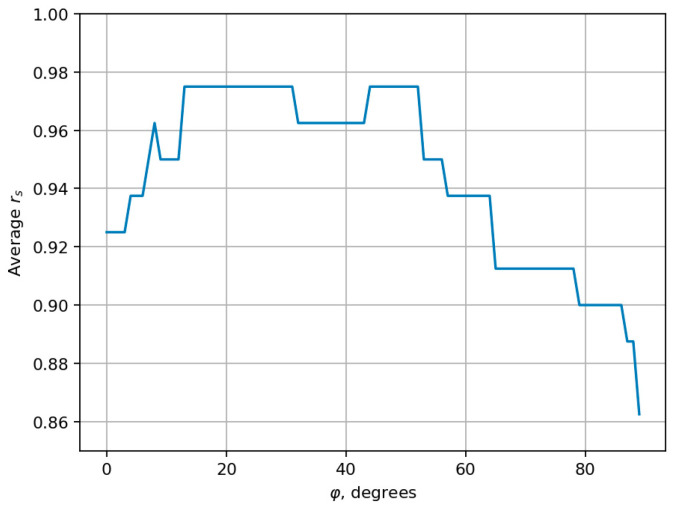
Dependency between the projection angle φ and the average Spearman’s rank correlation coefficient.

**Figure 7 sensors-25-06696-f007:**
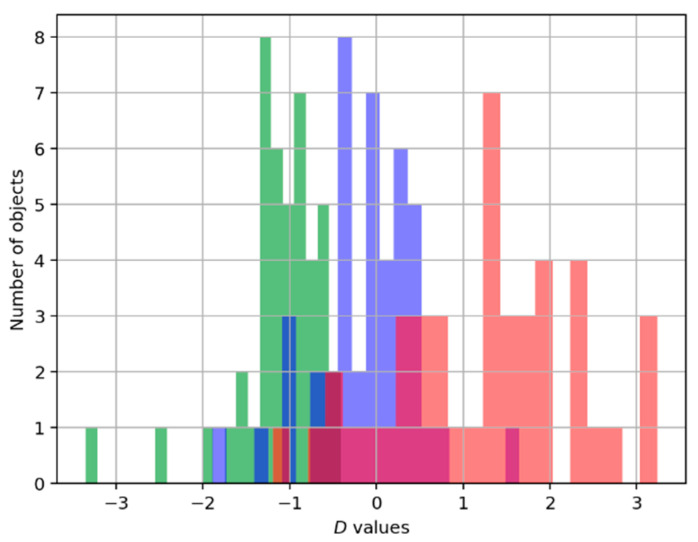
Diagram of projection of records to the axis under the angle of $13^{{}^{\circ}}$ in the dissimilarity space. Green means “good” execution, blue means “intermediate”, and red means “bad” execution of an exercise.

**Figure 8 sensors-25-06696-f008:**
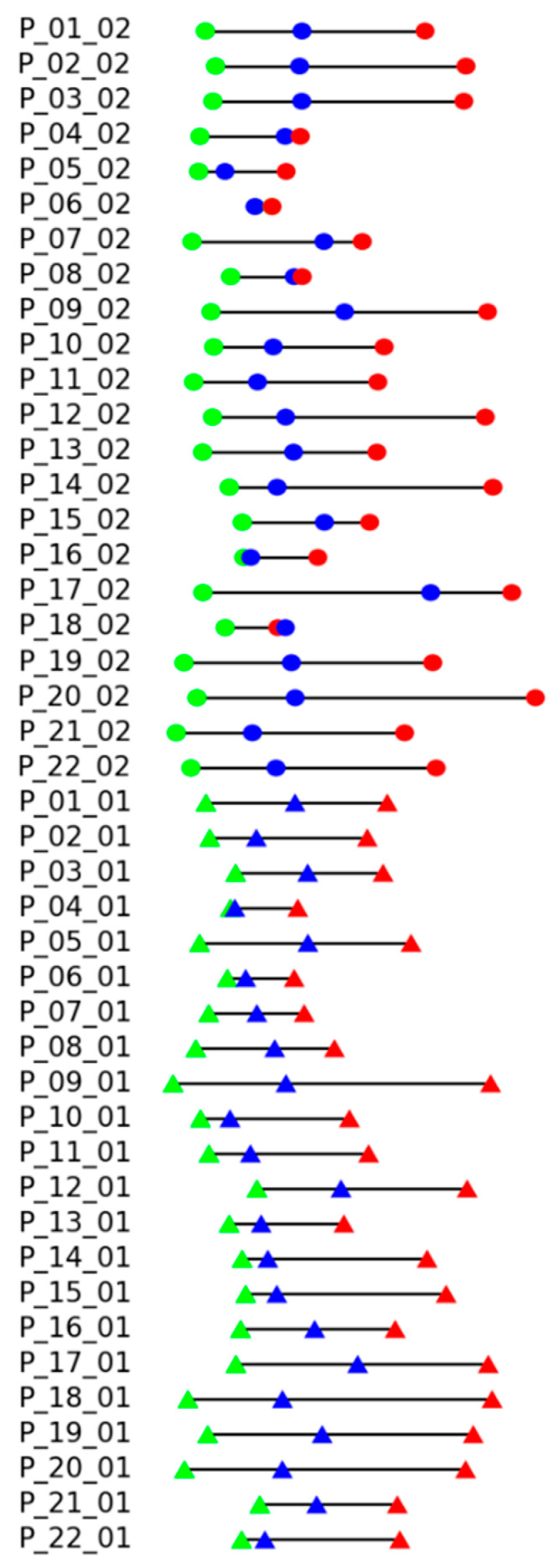
Experimental results: projections of all patients coordinate in *DP-DT* space into the line with the angle $\varphi =13^{{}^{\circ}}$. The color markers (dots for sitting, triangles for standing exercises) reflect the exercise execution levels: the green color for “good”, blue color for “intermediate”, and red color for “bad”.

**Figure 9 sensors-25-06696-f009:**
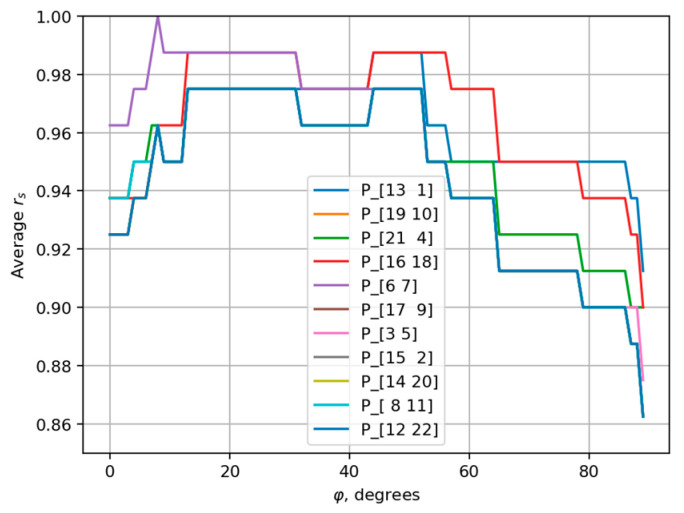
Dependency between the projection angle φ and the average Spearman’s rank correlation coefficient for 11 data subsets. The excluded patient IDs in each subset are shown in the legend in square brackets.

**Table 1 sensors-25-06696-t001:** Optimal balanced φ and corresponding average Spearman’s rank correlation coefficient in cross-validation procedure.

Excluded Patients	$\mathrm{Optimal}\;\varphi^{{}^{\circ}}$	$\mathrm{Average}\;r_{s}$
P_1, P_13	44	0.9875
P_10, P_19	13	0.975
P_4, P_21	13	0.975
P_16, P_18	13	0.9875
P_6, P_7	8	1.0
P_9, P_17	13	0.975
P_3, P_5	13	0.975
P_2, P_15	13	0.975
P_14, P_20	13	0.975
P_8, P_11	13	0.975
P_12, P_22	13	0.975

## Data Availability

The datasets used in the current study are available from the authors on reasonable request.

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
