# Peer review of "Automated Control of Rehabilitation Process in Physical Therapy Using a Novel Human Skeleton-Based Balanced Time Warping Algorithm"

_sensors, 2025, doi:10.3390/s25216696_

Round 1

Reviewer 1 Report

Comments and Suggestions for Authors

1.The dataset includes only 22 participants and 134 video sequences, which is a relatively small sample size. Although the experimental environment is well controlled, this remains insufficient to demonstrate the generalizability of the proposed method. In additonal,the authors should more explicitly explain why a self-collected dataset was chosen, as well as its advantages and limitations.

2.The classification results are only described qualitatively (clustering of “good,” “intermediate,” “bad” cases).Can there be more refined quantitative indicators.

3.The conclusion mainly reiterates the methodology without adequately emphasizing the key findings, the distinctive contributions compared with existing methods, or the potential significance for rehabilitation practice.

Comments on the Quality of English Language

Some sentences are too long, which makes the text less fluent to read.

Reviewer 2 Report

Comments and Suggestions for Authors
  • Why does the evaluation focus solely on Spearman’s rank correlation? This is a helpful metric for ranking, but it doesn’t fully reflect classification performance.
  • Please include additional statistical metrics such as confusion matrices or class-specific precision and recall to provide a more complete picture of model accuracy
  • How well does the system handle real-world variability? For example, do lighting conditions, clothing differences, or frame drops from RGB-D cameras affect skeletal extraction and pose comparison? Please add a robustness test across different recording conditions.
  • Please clarify how the skeleton standardization process handles unusual or incomplete poses, especially in patients with severe motor impairments. Any mechanism to detect and correct for missing joint data?
  • The concept of balancing posture and dynamic dissimilarity using a tunable angle (φ) is interesting. However, how sensitive is the system to this choice in unseen exercises or patient groups?
  • Whether the alignment algorithm supports streaming input? Can this be deployed for real-time monitoring or is it strictly for post-session analysis?
  • The experimental dataset is informative but limited to simulated impairments. Please conduct or add a follow-up study involving actual patients under medical supervision.
  • How many skeleton frames per second does the algorithm process?
  • The literature review is comprehensive but omits several recent and relevant works. In particular, the authors should consider citing the following research, which explores similar goals through different techniques such as gesture-based rehab systems, joint angle tracking, and sensor fusion. These papers could offer comparative insight into time-series processing and real-time feedback mechanisms in rehabilitation. Relevant DOI is: https://doi.org/10.1007/s10845-023-02152-x 

Reviewer 3 Report

Comments and Suggestions for Authors

The authors presented a dynamic time warping approach to assess the exercises in physical therapy, but the paper suffered two major shortcomings:

  1. The authors described the proposed method by citing reference papers of 23 to 26 without presenting the essential details of each step that makes the proposed approach difficult to comprehend.
  2. The literature review, method, results and discuss were mixed. For example, figure 4 was described before the Section 4. Experimental Study and Results; and in Section 4, the details of method and results were mixed. I would suggest the authors to restructure the paper to present the method and results better.

Minor issue is that "Criterion (1) and (4)" appeared in the lines 317 and 355 without explains.

Reviewer 4 Report

Comments and Suggestions for Authors

The paper sets out to develop a dynamic time-warping algorithm tailored to skeleton-based exercise performance evaluation. The differences between reference and patient movement are decomposed to pose-related and dynamics-related (time-related) components. A successful classification based on qualitative execution criteria is presented as evidence that these components are meaningful. Then, an algorithm is presented to unify these error components to arrive at a one-dimensional metric of dissimilarity. Overall, the work has merit, but the presentation of the methodology and the results need improvement and clarifications to accurately assess the work.

Abstract

The title and abstract highlight the control and evaluation of physical exercises. It is not evident how an assessment tool that gives a numerical/categorical feedback on execution quality in itself constitutes "control". Does the system provide specific prompts about improving exercise execution?

"The total dissimilarity measure is [...] optimized using Spearman's rank correlation". It is unclear what this refers to at this point, please elaborate.

"high correlation between computed dissimilarity and execution quality". Consider specifying the kind of correlation measure here since it is between a scalar and an ordinal variable.

1. Introduction

Some parts of the introduction are too general. Consider moving relevant paragraphs from the literature review here. It is also advisable to mention your own previous works that you are building on, for example the ones you cite regarding the methodologies. This helps the reader to grasp the broader direction of your research.

2. Literature Review

In general, the paragraphs are hard to parse out. Please improve the separation of the different approaches and studies that are presented.

Numerous different datasets are mentioned. It would be worthy to collect these into a table including the relevant references for those datasets.

At the end of the review section, instead of just repeating the contributions almost word for word in the introduction, provide rationale of why you chose these concrete approaches and modifications.

3. Proposed Methodology

The first two paragraphs belong more to the literature review as these provide information about related works.

Please clarify the source of Figure 1.

Overall, I found that the description of the methods needs improvement to make it easier to understand. It would help to include a brief overview at the beginning, where you first introduce "the general architecture of the system" (L246) and Figure 2. I was surprised when I was expecting to read more about the system and instead the description jumped to Figure 3.

It also surfaces only at L263 that you are building on your own previous paper (cited as [24]) regarding the dissimilarity measure evaluation. Why was this work not introduced earlier with the related works? Also in the caption of Figure 3, there is another reference (cited as [25]).

It was also somewhat surprising to find Figure 4 in this section as it clearly presents results from your current study's experiments.

4. Experimental Study and Results

The experiments are described adequately. However, the main presented results in Figure 5 and the corresponding description is hard to follow and possibly misleading. If the axes show difference in pose and dynamics, I would expect that the instructor's performance it at the origin, and good performance clusters close to the origin. It does not make sense to otherwise.

Also, the methodology for the classification itself is not described at all: L409-410: "Classification was carried out using Multi-layer Perceptron classifier with standard training parameters." Please expand and clarify.

Comments:

- Line 98: instead of 'unique', 'novel' is more appropriate

- typos: L106: 'resent' (recent)

- L336: "(see magenta highlights)": I did not find any such text or highlights.

- Figure 5: please also specify what the shaded area colors mean in the caption or as a figure legend.

Comments on the Quality of English Language

The English needs to be improved to make the content more clear. Some examples:

- L120-121: "found DTW warping from that." - this sentence makes no sense

- L252-254 also makes little sense

Round 2

Reviewer 2 Report

Comments and Suggestions for Authors

accept. Good Luck!

Reviewer 3 Report

Comments and Suggestions for Authors

The manuscript is structured better than previous version. I have no further comments.

Reviewer 4 Report

Comments and Suggestions for Authors

I have carefully reviewed the revised manuscript and the Authors' response letter. I am pleased to see that the manuscript improved significantly and the Authors addressed my concerns and comments adequately. I support the publication of the paper.